

# The majority are not performing home-exercises correctly two weeks after their initial instruction—an assessor-blinded study

Mathilde Faber[1], Malene H. Andersen[2], Claus Sevel[3], Kristian Thorborg[4], Thomas Bandholm[5] and Michael Rathleff[6,7]

[1] Kjellerup Fysioterapi og Træning, Kjellerup, Denmark
[2] Fysioterapeutisk Specialistteam, Aarhus, Denmark
[3] Faculty of Health Sciences, Physiotherapy, VIA University College, Aarhus, Denmark
[4] Sports Orthopedic Research Center—Copenhagen (SORC-C), Arthroscopic Centre Amager, Copenhagen University Hospital, Copenhagen, Denmark
[5] Physical Medicine & Rehabilitation Research—Copenhagen (PMR-C), Department of Occupational and Physical Therapy, Department of Orthopedic Surgery, and Clinical Research Center, Hvidovre Hospital, University of Copenhagen, Copenhagen, Denmark
[6] Department of Health Science and Technology, Center for Sensory-Motor Interaction (SMI), Aalborg University, Denmark
[7] Department of Occupational and Physiotherapy, Aalborg University Hospital, Aalborg, Denmark

Corresponding author
Michael Rathleff,
michaelrathleff@gmail.com

## ABSTRACT

**Introduction.** Time-under-tension (TUT) reflects time under load during strength training and is a proxy of the total exercise dose during strength training. The purpose of this study was to investigate if young participants are able to reproduce TUT and exercise form after two weeks of unsupervised exercises.

**Material and Methods.** The study was an assessor-blinded intervention study with 29 participants. After an initial instruction, all participants were instructed to perform two weeks of home-based unsupervised shoulder abduction exercises three times per week with an elastic exercise band. The participants were instructed in performing an exercise with a predefined TUT (3 s concentric; 2 s isometric; 3 s eccentric; 2 s break) corresponding to a total of 240 s of TUT during three sets of 10 repetitions. After completing two weeks of unsupervised home exercises, they returned for a follow-up assessment of TUT and exercise form while performing the shoulder abduction exercise. A stretch sensor attached to the elastic band was used to measure TUT at baseline and follow-up. A physiotherapist used a pre-defined clinical observation protocol to determine if participants used the correct exercise form.

**Results.** Fourteen of the 29 participants trained with the instructed TUT at follow-up (predefined target: 240 s ±8%). Thirteen of the 29 participants performed the shoulder abduction exercise with a correct exercise form. Seven of the 29 participants trained with the instructed TUT and exercise form at follow-up.

**Conclusion.** The majority of participants did not use the instructed TUT and exercise form at follow-up after two weeks of unsupervised exercises. These findings emphasize the importance of clear and specific home exercise instructions if participants are to follow the given exercise prescription regarding TUT and exercise

form as too many or too few exercise stimuli in relation to the initially prescribed amount of exercise most likely will provide a misinterpretation of the actual effect of any given specific home exercise intervention.

## INTRODUCTION

Elastic exercise bands are often used during rehabilitation of patients (*Alvarez et al., 2006*; *Andersen et al., 2010*; *Andersen et al., 2011*; *Jensen et al., 2014*; *Rathleff et al., 2014a*; *Thomas et al., 2005*). An elastic exercise band is a versatile training tool allowing patients to perform different types of home-based training including strength training and injury prevention training (*Alvarez et al., 2006*; *Andersen et al., 2011*; *Jensen et al., 2014*). Using elastic exercise bands during rehabilitation has multiple advantages; they are user-friendly, can be adjusted to different resistances, they are cheap and they do not take up much space (*Colado et al., 2010*; *Matheson et al., 2001*; *Melchiorri & Rainoldi, 2011*). Multiple studies have compared the effect of training with elastic exercise bands with the effect of training with free weights (e.g., dumbbells) on neuromuscular activation during exercise. These studies generally show that training with elastic exercise bands activates the contracting agonists to the same level as dumbbells, and therefore, the elastic exercise bands are expected to be just as efficacious in improving strength and reducing pain during a rehabilitation program. (*Andersen et al., 2010*; *Beers et al., 2008*; *Colado et al., 2010*; *Melchiorri & Rainoldi, 2011*).

In clinical practice, elastic band exercises are often used during home-based unsupervised training where the physiotherapist provides the patient with an initial instruction on how to perform the exercise. Included in these instructions are load, time under tension (TUT), range of motion (ROM), number of repetitions, sets, pauses between exercises, and the appropriate start position (*American College of Sports Medicine, 2009*; *Pereira & Gomes, 2003*). All these factors are important as they are closely linked to the total exercise dose and thus important for the clinical stimulus received by the patient. Especially TUT seems to play a large role in relation to the total exercise dose received during a training session (*American College of Sports Medicine, 2009*; *Andersen et al., 2011*; *Buitrago et al., 2012*; *Pereira & Gomes, 2003*; *Tran, Docherty & Behm, 2006*). Total TUT refers to the total time of all concentric, quasi-isometric and eccentric contraction-phases in a single training set (*Skovdal Rathleff, Thorborg & Bandholm, 2013*; *Tran, Docherty & Behm, 2006*). In combination with load and movement velocity it is an important strength training descriptor as it reflects the time factor of the strength training stimulus (*Buitrago et al., 2012*; *Gentil, Oliveira & Bottaro, 2006*; *Tran, Docherty & Behm, 2006*). Physiologically, a higher amount of TUT has been shown to increase myofibrillar protein synthesis more than a lower amount of TUT after a single, work-matched, strength training session in healthy subjects (*Burd et al., 2012*; *Tran & Docherty, 2006*). Hence, the quantification of the total TUT of performed strength training of the shoulder abductors is important in

order to determine if the executed training constitutes a sufficient clinical and physiological stimulus (*Buitrago et al., 2012*; *Burd et al., 2012*; *Gentil, Oliveira & Bottaro, 2006*; *Goto et al., 2009*; *Munn et al., 2005*; *Nogueira et al., 2009*; *Toigo & Boutellier, 2006*; *Tran & Docherty, 2006*; *Tran, Docherty & Behm, 2006*).

After a single initial instruction, the patients often complete their exercises unsupervised at home and return for follow-up assessment after, e.g., two weeks (*Andersen et al., 2011*; *Beers et al., 2008*; *Rathleff et al., 2014a*; *Thomas et al., 2005*). During the execution of the home exercises is it assumed that the patient follows the exercises as instructed and prescribed by the physiotherapist and thereby receives the exercise dose intended by the physiotherapist. However, the question remains if the patients are able to follow the exercise prescription and perform the exercise as instructed after only a single initial instruction as common in clinical practice. Clinical experience suggests that exercises are performed too fast at follow-ups thus reducing TUT. A correct execution of home-based exercises is vital in order to know the exercise dose received by the patients during their home-based unsupervised exercises. Exercise dosages that are either too small or too large in relation to the initially prescribed dosage introduce misinterpretation when concluding on effect estimates of any specific intervention in both clinical scenarios and in research.

Therefore, the purpose of this study was to investigate if young participants were able to reproduce TUT and exercise form after two weeks of unsupervised shoulder abduction. This exercise was selected as it has previously shown to be effective in reducing pain in the neck and shoulder muscles (*Andersen et al., 2011*; *Andersen et al., 2008b*; *Andersen et al., 2008a*; *Salo et al., 2010*; *Walther et al., 2004*). The main hypothesis was that participants would perform a shorter TUT at follow-up and thus not perform the exercise as instructed.

## METHODS

### Design

This study was designed as an assessor-blinded intervention study with baseline measurements and follow-up after 14 days of unsupervised home-based exercises.

### Ethical considerations

According to the Helsinki Declaration, all participants were provided with verbal and written information about the purpose of the study (*WMA Declaration of Helsinki, 2013*). They signed a written consent as part of the duty to the Ethics Committee (*Justitsministeriet, 2000*), and The Central Denmark Region Committee on Health Research Ethics approved the study (ref: 80/2014).

### Setting and participants

The study was conducted at VIA University College in Aarhus, Denmark, and at a hospital in Denmark. Young pain-free physiotherapy students from VIA University College, Aarhus, were recruited through advertisements at VIA University College, Aarhus. The participants were informed that the purpose of the study was to investigate whether physiotherapy students were capable of reproducing a shoulder exercise 14 days after their initial exercise instruction. The information to the participants was kept short in order to

blind them for our primary outcome (TUT). Inclusion criteria included fully active range of motion (AROM) in the shoulder joint and no pain in the shoulder region.

## Equipment

A stretch sensor was used to measure TUT at baseline and follow-up but not during the 14 days of unsupervised home-based exercise. The stretch sensor was based on the technology designed by Danfoss PolyPower. It acted as an elastic capacitive material stretchable in one direction. It allowed us to measure how much the sensor was being stretched (for further details refer to *Kappel et al. (2012)*). The sensor was attached to the rubber band through two clips that made the sensor easily transferable to other elastic exercise bands. The sensor was attached via a USB to a small box that recorded data 200 times per second. A switch was mounted in the handle of the elastic exercise band so that the data recording would start whenever the handle was pressed. The participants were told that the sensor could measure if the exercise was performed correctly. They were not informed about the primary purpose of measuring TUT.

The elastic bands were standard red and green Thera-Bands, which are commonly used in rehabilitation studies (*Andersen et al., 2008a*; *Andersen et al., 2008b*; *Andersen et al., 2011*; *Salo et al., 2010*; *Walther et al., 2004*). To ensure that each strength exercise scenario was performed with the same external resistance, corresponding to a relative load of 12 repetition maximum (RM), lines were drawn for every 5 cm in the entire length of the exercise band. The 12 RM load was determined prior to data collection and reflected a load typically prescribed to treat conditions of the shoulder and neck (*Andersen et al., 2008a*; *Andersen et al., 2008b*; *Andersen et al., 2011*; *Salo et al., 2010*; *Walther et al., 2004*).

## Test procedure

The participants were instructed to perform a shoulder abduction exercise using a Thera-Band. This specific exercise has proven effective in reducing pain in the neck and shoulder muscles (*Andersen et al., 2008a*; *Andersen et al., 2008b*; *Andersen et al., 2011*; *Salo et al., 2010*; *Walther et al., 2004*), and this type of live instruction has previously shown to be effective in learning exercises and reflects clinical practice (*Reo & Mercer, 2004*). The instruction was to stand with hip width distance between the feet and place one end of the elastic band under the foot and the other in the hand. Then the subject was to raise the arm slightly in front of the body from 0 to 90 degrees shoulder abduction and 30 degrees horizontal shoulder flexion. The elbow was to be in a slightly flexed position during the entire range of motion. The palm was to face the floor. The participants were instructed that tension was to be present in the elastic band at 0 degrees' abduction (see Fig. 1).

Further, they were instructed that each exercise session should consist of three sets of 10 repetitions at 12 RM with a two-minute break between each set. A mark was made on the elastic band at a length corresponding to 12 RM. TUT was 3 s concentric (con); 2 s isometric at 90 degrees' abduction; 3 s eccentric (ecc); 2 s pause at 0 degrees' abduction. Therefore, total TUT during the three sets of 10 repetitions was 240 s.

During the initial instruction, the participants received clear and thorough instructions on how to perform the shoulder abduction exercise using the prescribed TUT and the

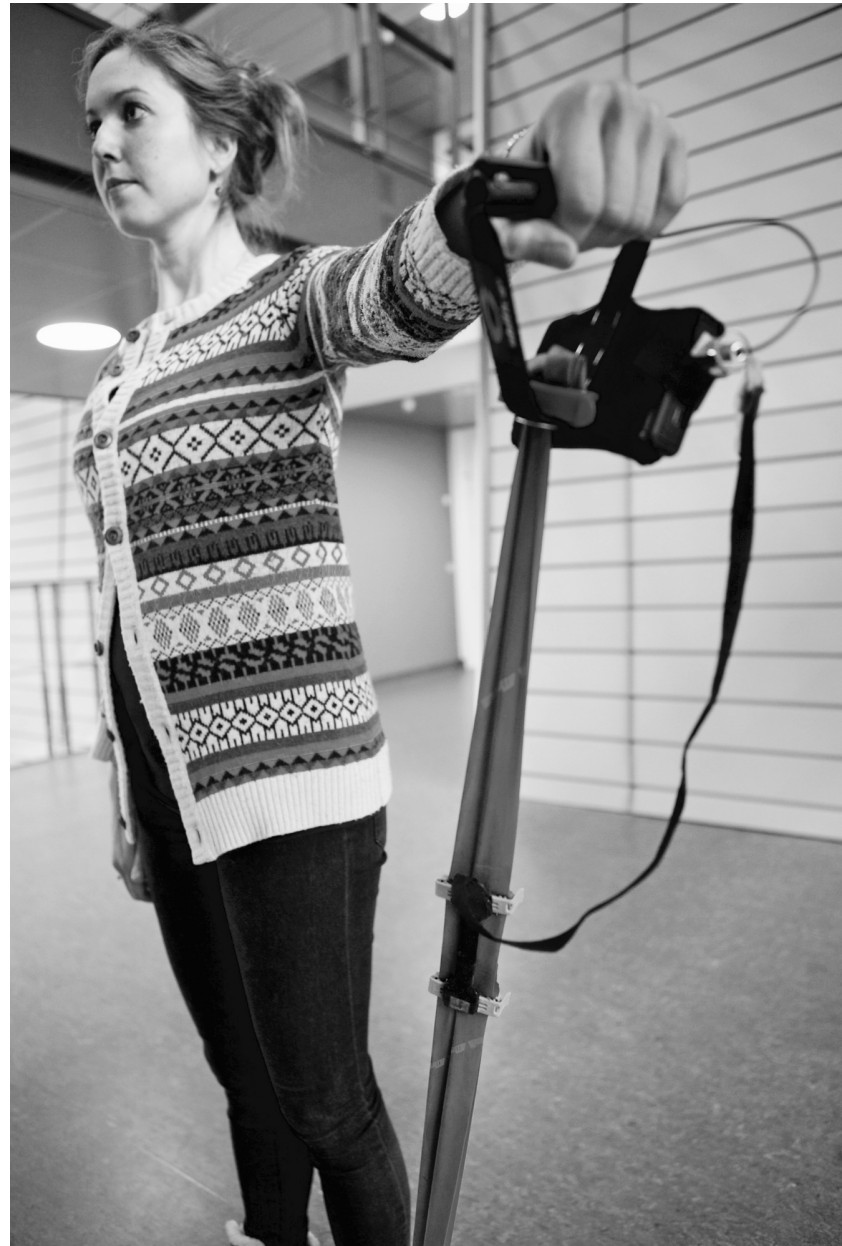

**Figure 1  Shoulder abduction with the stretch-sensor attached to the elastic band.**

correct exercise form. First, the participants saw the assessor perform the exercise correctly, and afterwards the participants performed the exercise themselves with feedback from a metronome to guide the correct TUT. The participants were given up to five attempts to find the correct TUT by following the beat of the metronome. After the practice, they had a two-minute break before the data collection. Each participant then completed 3 sets of 10 repetitions, with a two-minute beak between the sets. The participants were verbally guided through the test if they did not follow the beat of the metronome, or if they did not use correct exercise form. Therefore, all participants performed the exercise correctly at

baseline. After the exercise, the participants were given a training diary as well as date and time for the follow-up. They were informed that the exercise had to be performed at home three times every week for two weeks. The exercise instruction lasted 7–10 min and resembled the time spent on exercise instructions for one exercise in normal clinical practice.

Fourteen days later the participants returned for the follow-up assessment and were asked to perform three sets of 10 repetitions, using the exercise form as instructed at the first visit and practised at home for the past two weeks. At the follow-up assessment no verbal feedback was given and the metronome was not turned on. The participants were also asked to hand over their training diary to the assessor.

## Outcomes

The primary outcome was defined as the total TUT during all three exercise sets with the aim to investigate if the participants were capable of reproducing total TUT during the shoulder abduction exercise. Matlab (r2011a; MathWorks, Nattick, USA) was used to analyse the data from the stretch-sensor. A custom-written Matlab programme transformed the stretch sensor data into an image showing the stretch of the sensor as a function of time. The rating of every repetition was done manually by moving the mouse cursor to the visually observed time points corresponding to the contraction-specific phases. This approach has previously been used and demonstrates good validity and reliability (*Skovdal Rathleff, Thorborg & Bandholm, 2013*).

The secondary outcome was the exercise form. This was measured using a clinical observation form similar to a previous study (*Reo & Mercer, 2004*). The following parameters were noted: hip width distance between the feet, 0–90 degrees shoulder abduction, 30 degrees horizontal flexion, palm facing the floor, slight flexion of the elbow, tension in the elastic exercise band at 0 degrees' shoulder abduction and alignment though the standing position. The assessor noted the number of errors in the exercise execution by visual observation.

## Rating of data

The assessor was a female physiotherapist with no previous experience in rating data from the stretch sensor. The assessor received one hour of practice rating data from the stretch sensor using data from participants who were not part of the current study. The rating of data was carried out after completion of all measurements from baseline to follow-up. After this the collected data from each participant, from both baseline and follow-up, were anonymised and given a random number. The assessor was blinded and therefore, did not know whether the data were from baseline or follow-up. Further, the assessor did not know which of the three exercise sets she was rating.

## Intertester reliability

Before commencing the current study, a pilot study was performed where two independent participants performed six sets of 10 repetitions. Two independent assessors rated the stretch sensor data. This analysis was performed to determine the interrater reliability and an estimate of the worst-case scenario of reliability (it was expected that intrarater

reliability would be higher). The intraclass correlation coefficient (ICC 2.1) was used as a relative expression of the reliability while limits of agreement (LoA) were used as an absolute expression of agreement. The ICC for the interrater reliability for single TUT was 0.93. The agreement between raters for TUT ranged from −0.57 to 0.56 s (8% of mean TUT). Therefore, the predefined target for total TUT was 240 s ±8% (limits: 220.8–259.2 s) to reflect measurement uncertainty.

## Sample size

As the main hypothesis was that the participants would use a shorter TUT at follow-up, the study was powered to detect a 20-s lower TUT (approximately 8% of the mean TUT) at follow-up compared with baseline. Using a standard deviation of 25 s at 5% significance and 80% power, it was necessary to include 25 participants. To account for potential dropouts, we included 32 participants.

## Statistical analysis

All calculations were performed using Stata version 11 (StataCorp, College Station, Texas, USA). Descriptive statistics are presented as mean and standard deviation. The level of statistical significance was set as $P < 0.05$. Parametric statistics were used because the data were normally distributed. Total TUT from baseline to follow-up was compared using paired $t$-tests.

## RESULTS

A total of 35 participants signed up for the study, but three of them had shoulder pain and did not fulfil the inclusion criteria and were excluded from the study. A total of 32 physiotherapy students aged 20–27 (10 men and 19 women) fulfilled the inclusion criteria and were included. Further three participants dropped out during the two weeks of unsupervised exercises: Two sustained an injury to the arm (unrelated to the exercise) and could not participate, and one did not turn up for follow-up for unknown reasons. Therefore, 29 participants completed both baseline and follow-up visits.

Twenty of the 29 participants completed the six exercise sessions over two weeks as prescribed. One participant completed the exercise more than the recommended 6 times, and 8 completed the exercise less than recommended. One participant completed the exercise three times, two participants completed the exercise four times and five participants completed the exercise five times during the 14 days.

Total TUT at baseline was 255.2 s (±10.6). At follow-up, total TUT was 252.5 s (±41.0 s) with a range from 160 s and up to 370 s. No significant difference was detected in total TUT between baseline and follow-up (2.6 s 95% CI[−13.2; 18.5]) $p = 0.74$. Fourteen of the 29 participants used the instructed TUT at follow-up (predefined target: 240 s ±8%). Six used a lower TUT and 15 used a higher TUT than instructed (Fig. 2).

Thirteen of the 29 participants performed the shoulder abduction with the correct exercise form. Seven of the 29 used both the instructed TUT and exercise form at follow-up. The most common mistakes were: no flexed position in the elbow and abducting the arm to more than 90 degrees (Table 1).

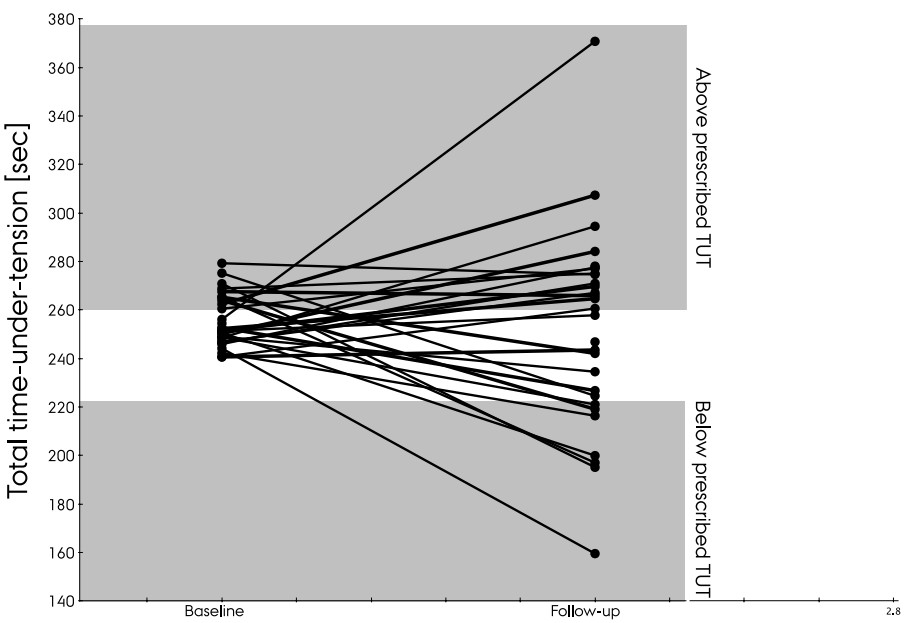

**Figure 2** Total time-under-tension at baseline and follow-up.

**Table 1** Exercise form at follow-up.

| Exercise parameters | Number performing the exercise as instructed |
|---|---|
| Hip width distance between the feet | 26/29 |
| 0–90 degrees shoulder abduction | 24/29 |
| 30 degrees horizontal flexion | 25/29 |
| Palm facing the floor | 28/29 |
| Slight flexion of the elbow | 20/29 |
| Tension in the elastic exercise band at 0 degrees shoulder abduction | 28/29 |
| Alignment though the standing position | 29/29 |

## DISCUSSION

Contrary to our hypothesis, the participants did not systematically use a lower TUT at follow-up but varied considerably. Half of the participants used the instructed TUT and less than half used the instructed exercise form at follow-up. Physiotherapists may think they know the exercise dose received by the patients during their home-based exercises, but these results suggest that some receive a higher exercise dose, while others receive an exercise dose too low to constitute a sufficient clinical stimulus.

### Practical relevance of the study

At baseline the participants had a mean total TUT of 255.2 s with a standard deviation of 10.6 s. At follow-up no significant change appeared in mean TUT, but the standard deviation increased four-fold to 41.0 s with a range from 160 s and up to 370 s. If TUT at

follow-up reflected the exercise dose during unsupervised exercises, then some participants would receive a 50% higher exercise dose than instructed, while others would receive a 40% lower dose. The largest concern would most likely be those receiving a lower exercise dose than intended as it may not elicit a sufficient clinical stimulus. Participants receiving an exercise dose higher than prescribed may end up with a secondary overuse injury hampering the rehabilitation of their original injury. However, the main concern with patients not receiving the prescribed exercise dosage is that decisions on further progression or cessation of a specific program are difficult. It leaves the treating therapist or researcher in doubt whether the lack of progression or the increase in symptoms is due to not following the exercise intervention as prescribed, or whether this is caused by other factors than the actual intervention.

The large difference in exercise dose may result in different clinical outcomes (*Osteras & Torstensen, 2010*; *Rathleff et al., 2015a*). Previous studies suggest that the length of TUT influences the physiological response, and longer TUT is associated with larger physiological response (*Burd et al., 2012*; *Munn et al., 2005*; *Tran & Docherty, 2006*). The optimal TUT may depend on the goal of the exercise and the response of the patients (*American College of Sports Medicine, 2009*; *Carpinelli, Otto & Winnett, 2004*). Nevertheless, TUT needs to be defined when giving exercise prescriptions in order to describe the exercise dose (*Fukumoto et al., 2014*; *McBride et al., 2009*; *Toigo & Boutellier, 2006*; *Tran & Docherty, 2006*).

At baseline the participants received a live instruction of the exercise by a physiotherapist. This approach has previously been shown to improve the performance of an exercise compared to a written leaflet, both immediately after the instruction and after one day (*Reo & Mercer, 2004*). Both the objective and subjective outcome of this study demonstrate that it is difficult for participants to perform the exercise as instructed. These findings support older studies using subjective outcomes to determine if the exercise was performed as instructed and highlight how difficult it can be to perform an exercise correctly after a short initial instruction (*Reo & Mercer, 2004*; *Henry, Rosemond & Eckert, 1999*). This raises the question of how thorough an initial instruction is to be before the physiotherapist can send off the patient to perform home-based exercises. One way to ensure proper exercise execution would be to perform the exercise under supervision during the first couple of weeks and then slowly transit to home-based exercises. Based on these results, it is assumed that some kind of feedback may be needed to avoid a too low exercise dose. This could be done using a simple metronome on the mobile phone of the patient or using exercise-integrated tools such as the Bandcizer to continuously monitor TUT and adherence to exercise (*Rathleff et al., 2015b*). Indeed, such information collected during every home-based exercise session would help qualify the follow-up visits where a patient is not responding as expected based on the treatment plan.

## Strengths and weaknesses of the study

The primary outcome, TUT, has previously been shown to be valid and highly reliable and the participants were blinded to the primary outcome (*Skovdal Rathleff, Thorborg &*

*Bandholm, 2013*). The assessor who rated data was blinded towards the rating of baseline or follow-up data. The assessor judging the exercise form was not blinded, which may have biased the secondary analysis on the exercise form. No TUT data were measured during the home-based exercises, and it is unknown if the subjects performed their exercise differently while at home. The participants were all physiotherapy students and aware that they were being watched while performing the exercise. Further, it might be speculated that physiotherapy students are better at understanding exercise instructions and thereby have a less steep learning curve compared with, e.g., patients. Therefore, these results are most likely a conservative estimate and it would be expected that an even lower proportion of the patients in clinical practice will perform the exercises using the prescribed TUT and exercise form. It is also possible that physiotherapy students would have less motivation to perform the exercises as instructed than patients because patients suffer from pain and functional limitation which could motivate them to do the exercises as instructed.

### Future research

Future studies should investigate if the current findings also apply to patients with shoulder disorders. There is a need to investigate the amount and type of instruction required in order for the participants to perform the exercise as instructed or if real-time feedback during exercises is required. To further elucidate why the participants are not able to reproduce the exercise, an interview could be performed asking the participants questions on this issue. This could provide valuable information on participant preferences for type and form of the exercise instruction.

## CONCLUSION

Contrary to our hypothesis, the participants did not systematically perform a lower TUT at follow-up but varied considerably, both above and below the instructed TUT. Less than 25% of the participants performed the instructed TUT and the correct exercise form at follow-up after two weeks of unsupervised home-based exercises. These findings emphasize the importance of clear and specific home exercise instruction if participants are to follow the given exercise prescription regarding TUT and exercise form as too many or too few exercise stimuli in relation to the amount of exercise initially prescribed most likely will provide a misinterpretation of the actual effect of any given specific home-based exercise intervention.

### Funding

There was no external funding for this study.

### Competing Interests

Mathilde M.F. Faber is an employee of Kjellerup Fysioterapi og Træning and Malene H. Andersen is an employee of Fysioterapeutisk Specialistteam Aarhus.

## Author Contributions

- Mathilde Faber and Malene H. Andersen conceived and designed the experiments, performed the experiments, analyzed the data, contributed reagents/materials/analysis tools, wrote the paper, prepared figures and/or tables, reviewed drafts of the paper.
- Claus Sevel, Kristian Thorborg and Thomas Bandholm conceived and designed the experiments, contributed reagents/materials/analysis tools, wrote the paper, reviewed drafts of the paper.
- Michael Rathleff conceived and designed the experiments, performed the experiments, analyzed the data, contributed reagents/materials/analysis tools, prepared figures and/or tables, reviewed drafts of the paper.

## Human Ethics

The following information was supplied relating to ethical approvals (i.e., approving body and any reference numbers):

The Ethics Committee of the Central Region Denmark was contacted before the beginning of the intervention and approved the study.

## Supplemental Information

Supplemental information for this article can be found online at http://dx.doi.org/10.7717/peerj.1102#supplemental-information.

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
