# Peer review of "The majority are not performing home-exercises correctly two weeks after their initial instruction—an assessor-blinded study"

_PeerJ, doi:10.7717/peerj.1102_

## Round 0.1 · original submission · Major Revisions

Further to your Appeal, I am in agreement that this work is very premature but in principle sound. Therefore, the decision has been changed to 'Major Revisions' - please address the reviewer comments accordingly.

·

Basic reporting

Basic reporting requieres the following improvements:
1. the numurous language errors (title included) need to be corrected, preferrably by a native English speeking corrector.
2. The reason why this study has been performed should be clarified. What is the clinical problem to be solved? Please provide arguments for the hypothesis that patients would use shorter TUT or do not perform the exercises conform instructions after 2 weeks. Is the hypothesis based on findings in clinical practice? Are home-performed TUT exercises less effective than expected? What is described in the literature about studies investigating efficacy of shoulder exercises by elastic bands. How have the studies been performed? Were those exercises all performed under supervision of the physiotherapist? Has the efficacy of home-performed shoulder exercise with elastic bands been studied before?
3. Inclusion/exclusion criteria: Line 139: The part “A total of … visit” belongs to the results section. Please describe (in the results section) in what way particularly the three participants failed the inclusion criteria. I advise to add the first line of this alinea to the previous section “Setting an participants”.
4. Equipment: Line 148: Please describe clearly whether the elastic band with the stretch sensor is used by all participants at home or only during the measurements at baseline and follow-up. I assume the latter.
5. Lines 307-310: This belongs in the results section. What we need here is an interpretation of the results and the consequences for patients with shoulder problems.
6. In lines 312-317 the authors state that using less TUT is a concern, not so much using higher TUT. Can using higher TUT be harmful? Or do we only need to focus on improving instruction in order to prevent less TUT in exercise therapy? The authors need to be very specific here in order to develop better ways to instruct or supervise patients receiving home-based exercise therapy.

Experimental design

1. The experimental design is adequate, however and important question arises that could have easliy been adressed in this study. Unfortunately the question why the subjects were not able to reproduce the instructed exercise is not adressed. This knowledge may help to develop means that help patients to perform their exercises in the correct way. Therefore, if possible, I suggest to perform interviews adressing the reasons behind the failure of reproducing the exercisesand following instructions properly and tips for improvements.
2. In the introduction the hypothesis is stated, in the last part of the methods section the outcome paramaters are discribed. However, the research question is not clearly stated.
3. Line 271: the first result described is not mentioned as outcome parameter. I suggest to add this as a secondary outcome in het Methods section.

Validity of the findings

1. Line 298: I do not agree with the conclusion that the findings in this study are in contrast with the hypothesis. Some patients did indeed use less TUT at follow-up, but other used more. The main finding therefore is that only 7 out of 29 subjects performed the exercise the proper way after 14 days at follow-up. This is an important an very disturbing finding and cannot be emphasized enough! The hypothesis has become irrelevant to the message that has to be sent in this paper.
2. Line 337: the authors state that feedback information will help. Is this an assumption or a finding (from literature)? It is a pity that, as said here above, that the subjects were not asked why they were not able to reproduce the exercise after 2 weeks. This could then be discussed in this section.
3. Line 349: Perhaps the physiotherapy students cannot be completely compared with patients with shoulder pain. Perhaps they are less motivated to perform the exercises in the correct way as they do not suffer from shoulder pain. However, pain might lead to less performance as exercise may lead to more pain during exercise. Moreover, one can envision that it is easier to explain exercises to physiotherapy students than to patients, resulting in better performance of physiotherapy students.
4. Line 359: This study certainly needs to be performed in patients with shoulder problems, after improving the instructiosn using the knowledge gained in this study. This should be enriched by asking the subjects about why they failed to reproduce the exercises and tips on how to improve instruction before and during exercising at home. The new instruction and support methods should then be tested in healthy subjects again and subsequently after in patients with shoulder problems.

Additional comments

This study reveals the difficulty to follow exercise instructions in an unsupervised environment after (thorough) instruction. This is an important finding that requieres indepth study and may lead the way to developing tools to support and improve home-performed exercises. This is important, since self management is an international trent and needs to be improved to be effective. This study is a very good start, but unfortunately the authors did not expand this study by interviewing the subjects to gain indepth information about why it was so difficult to repeat exercises after 2 weeks after instruction. Then specific recommendations can be made to improve intructions for home-performed exercise theray and tools can be developed to improve and support thi. as such self management and subsequently patient health and recovery is improved.

Reviewer 2 ·

Basic reporting

There are some minor language issues:
line 67 during rehabilitating > during rehabilitation
line 101 returns > return
line 301-304 "the physiotherapist may think.." > this sentence should be formulated differently, multiple language issues
line 335 too > to

Experimental design

No comments.

Validity of the findings

The results section is very short, with only 1 figure. Baseline characteristics table (Table 1) is missing, with gender, age, BMI, education level etc. Therefore, possible confounding is not assessed. In the section "strengths and weaknesses of the study", it is stated that physiotherapy students have been watched and so might perform better, or in contrary, could perform worse than patients due to lack of pain and motivation. However, another issue is that physiotherapy students should better understand instructions than an average patient, especially if physiotherapy student is almost graduating and becoming a physiotherapist. This possible confounder was not assessed.
There should be also a Table 2 with the parameters of secondary outcome (exercise form). It would be interesting to see a difference between all parameters at baseline and at follow-up.
In discussion, it is important to state whether higher dose of TUT is harmful of beneficial; if beneficial, than only those who receive lower TUT are a problem.
Also, it would be helpful to the reader if you compare the learning curve of your study participants with other comparable studies (students? healthy volunteers? patients?)

Additional comments

With pleasure I read your article. I have some major (see section Validity) and minor (Basic reporting) comments.

---

## Round 0.2 · accepted · Accept

Although still considered by me as a preliminary study, it has been clearly improved by the reviewers comments.

·

Basic reporting

The article meets the standards. It is now comprehensive and clearly written.
I advise to have the final version of this article checked again for possible errors in English language.
I doubt about "is to be" in this line in the discussion: "This raises the question of how thorough an initial instruction is to be before the physiotherapist can send off the patient to perform home-based exercises." My formulation would have been "must be".
If this line is approved for its language and no other changes in the text have been made, the article may be accepted as is.

Experimental design

No commensts

Validity of the findings

No comments

Additional comments

The objectve of the study was if subjects were capabale of reproducing home based exercizes after intstruction by the physiotherapist.The clear message is that less than half of the subjects are actually capable of reproding the exercizes after 2 weeks as instructed.The authors have well stated the clincical relevance of this finding and commented on further steps to solve this problem by further research and development of tools or methods to improve reproduction of instructed home-based exercize.

Reviewer 2 ·

Basic reporting

No comments.

Experimental design

No comments

Validity of the findings

Concerning my first comment on absence of Table 1 with baseline characteristics, possible confounding is still not assessed. Authors do not try to explain the different TUT between students by their gender, age, or education level (1st year student or 2nd year student, or last year student etc). So if analyzing men and women separately, for example, is there any difference in TUT at follow-up? Or is age of the student a predictor of different performance of TUT? I still believe that the author should at least try to use these available data to see if these confounders could influence the data. And if the study is not powered enough to answer these questions, it should be mentioned in the discussion. It is very important to know that the students mostly did not perform the right TUT, but it is also important to answer the question why, as this would lead to better design of the future investigation with patients.

Additional comments

I am satisfied with most answers, however, I still have a concern about assessment of confounders, see validity section.